# Evaluating teachers' pedagogical content knowledge in implementing classroom-based assessment: A case study among esl secondary school teachers in Selangor, Malaysia

**Rafiza Abdul Razak**[1], **Shahazwan Mat Yusoff**[1]*, **Chin Hai Leng**[1], **Anwar Farhan Mohamadd Marzaini**[2]

**1** Department of Curriculum & Instructional Technology, Universiti Malaya, Kuala Lumpur, Malaysia,
**2** Academy of Language Studies, University Teknologi MARA, Pulau Pinag, Malaysia

* shahazwan92@gmail.com

## Abstract

The Malaysian Education Blueprint (PPPM) 2013–2025 has spurred significant reforms in the Primary School Standard Curriculum (KSSR) and Secondary School Standard Curriculum (KSSM), particularly concerning classroom-based assessment (CBA). CBA evaluates students' understanding and progress, informs instruction, and enhances the learning outcomes. Teachers with robust pedagogical content knowledge (PCK) are better equipped to design and implement effective CBA strategies that accurately assess students' comprehension and growth, provide personalised feedback, and guide instruction. This study aims to investigate the relationship between PCK and CBA among English as a Second Language (ESL) secondary school teachers in Selangor, Malaysia. A 5-point Likert-scale questionnaire was administered to 338 teachers across 27 regional secondary schools in Selangor. The Covariance-based structural equation modelling (SEM) was used to analyse the data. The findings revealed that the secondary school teachers demonstrated a high level of PCK, with content knowledge (CK) obtaining the highest mean, followed by pedagogical knowledge (PK) and pedagogical content knowledge (PCK). The CBA practices among these teachers were also found to be high. SEM analysis showed a positive association between PK and CBA practices and between PCK and CBA. However, no positive association was observed between CK and CBA practices. In order to enhance teachers' PCK and ensure the effective implementation of CBA, which is crucial for student learning outcomes in Malaysian ESL secondary schools, it is recommended that continuous professional development opportunities be provided, specifically focusing on PCK and CBA.

**Data Availability Statement:** The dataset can be accessed through this link https://figshare.com/s/7d3979c34c3b844d558e.

**Funding:** The authors received no specific funding for this work.

# Introduction

The role of teachers in assessing student learning and development has evolved over the years, with a growing emphasis on incorporating classroom-based assessments (CBAs) into the educational process. CBA aims to create a more enjoyable and meaningful learning experience for students and it is viewed as an integral components of student-centred and inclusive education practices. In Malaysia, the implementation of the second phase of the Malaysian Education Blueprint (PPPM) 2013–2025 has led to notable transformations in classroom evaluation. This shift has moved the role of assessment from the exam-centric culture to being primarily managed by teachers, who now utilize the School-Based English Language Curriculum (DSKP) along with guidance from the Classroom-Based Assessment (CBA) handbook [1]. In this sense, Stiggins [2] accounted that the successful application of CBAs relies significantly on teachers' expertise in pedagogical content knowledge (PCK), content knowledge (CK), and pedagogical knowledge (PK).

Despite incorporating CBA in the Malaysian educational system, researchers have found that some English as a Second Language (ESL) secondary school teachers need help to implement CBA in their daily teaching practices [3]. The problem of inadequate knowledge and ineffective assessment practices among teachers has raised concerns over the potential impact on student's learning outcomes and overall educational quality. A comprehensive understanding of the relationship between teachers' PCK, CK, and PK and their CBA practices is crucial for devising strategies and policies to improve the implementation of CBAs in Malaysian ESL secondary school contexts.

Jones & Moreland [4] emphasised the importance of PCK, PK, and CK for English teachers working with students, as these competencies allow them to create engaging and effective educational experiences. In addition, Mat Yusoff et al. [5] highlighted the value of understanding language acquisition theories and applying them in practice. Building on these ideas, exploring the role of ESL teachers' Pedagogical Content Knowledge (PCK) in classroom-based assessment is crucial, which refers to integrating PCK, PK, and CK in real-world educational settings. In ESL education, classroom-based assessment is pivotal in providing teachers with meaningful information on students' language development and learning progress. Teachers must be knowledgeable in the subject matter and possess the skills to design and implement appropriate assessment strategies that cater to the unique needs of language learners. This is particularly significant, given that assessments can profoundly impact students' motivation, self-efficacy, and overall learning experience.

While studies emphasising the importance of PCK, Pedagogical Knowledge (PK), and Content Knowledge (CK) in ESL education have gained popularity in academic circles, there were dearth of studies which specifically investigates the association between PCK and the execution of CBA. The previous literature has limit their discussion in levelling these elements in ESL instruction [5–7], and understanding the association of beliefs and knowledge into PCK strategies [8]. Hence, the present study aims to fill the knowledge gap in the literature by examining the impact of ESL teachers' PCK on the quality of CBA implementation. This research will shed light on the relationship between teachers' pedagogical knowledge and their ability to design and implement effective assessment practices in the ESL classroom. Additionally, this study intended to identify teachers' challenges in enacting the new assessment system and propose potential strategies in improving assessment practices. The findings of this study lead to the practical implications in ESL instruction. Understanding how PCK influences assessment quality can inform the development of targeted professional development programs for teachers, focusing on enhancing their pedagogical knowledge in the assessment domain. This, in turn, can lead to more accurate and meaningful assessments, thereby improving the overall

educational experiences of young language learners. Thus, the research questions in this study are formulated as below;

Research questions:

1. What is the relationship between ESL teachers' content knowledge and classroom assessment practices among secondary school teachers?

2. What is the relationship between ESL teachers' pedagogical knowledge and classroom-based assessment practices among secondary school teachers?

3. What is the relationship between ESL teachers' pedagogical content knowledge and classroom-based assessment practices among secondary school teachers?

## Literature review

### The classroom-based assessment

Various studies related to classroom assessment utilised different terms to describe the procedure for monitoring the students' progress. The terms "approach," "tools and instruments," and "assessment procedures" have all been used interchangeably [9–12]. Teachers use various exams to give students the chance to succeed and exhibit their skills. Contrary to large-scale standardised testing, which frequently uses complex psychometric techniques, classroom assessment allows teachers to choose the best assessment strategy to acquire precise information on students' learning progress.

Teachers utilise various assessment techniques to gauge their students' learning, depending on the learning context and how well the technique fits the needs of obtaining accurate data. Researchers and educators have been researching the numerous student information-gathering techniques teachers use for decades. A growing corpus of research offers helpful insights into teachers' strategies [13–15].

A diverse range of assessment methods is deemed crucial for enhancing learning as it enables teachers to gather multiple sources of evidence that can comprehensively understand their students' progress towards their learning objectives [10]. According to Butler & McMunn [10], information about students' progress can be obtained through (1) General methods, including teacher-student interactions, questioning, and observation, and (2) Specific methods, such as oral presentations, written essays, or multiple-choice exams. These specific assessment methods are usually categorised into two broad categories, which may vary in nomenclature based on the researcher's preference.

In academic literature, the words "conventional assessment" and "alternative assessment" are frequently used to distinguish between closed-ended and open-ended questions, respectively. Multiple-choice, true/false, match, and other "chosen response" items are frequently seen in traditional exams [16]. The term "paper-and-pencil evaluations" is occasionally used to describe these tests [12]. However, "performance-based exams," which include oral presentations and compositional exercises like poem writing and recitation, are a kind of alternative assessment that permits a show of learning [16].

Alternative assessment is frequently used to evaluate students' knowledge exhibited through real-world tasks and activities. Unlike conventional paper-and-pencil tests, which typically measure verbal ability, alternative evaluations include pupils in active displays of their knowledge. Alternative assessments are considered significant for offering a more thorough evaluation of student's learning while being subjective, and they are frequently used in conjunction with standardised examinations to increase objectivity and fairness in assessments [17].

There has been some debate among researchers regarding the effectiveness of paper and pencil assessments in capturing crucial aspects of student learning. Some authors argue that such assessments must be improved in assessing higher-order skills like generative thinking, collaboration, and sustained effort over time [18]. However, the findings of a study by Hawes et al. [19] on the relationship between kindergarten students' symbolic number comparison abilities and future mathematics achievement suggest that paper and pencil assessments can be a valid tool for assessing early math skills. In contrast, performance-based assessments, such as portfolios, computer simulations, oral presentations, and projects requiring academic background, teamwork, and problem-solving abilities, have been touted as a better means of evaluating these skills [18].

In recent research, the term "paper and pencil-based assessment" has been used to describe a form of evaluation that consists of closed-ended and more structured questions that are usually administered on paper. Such assessments include objective evaluations where students are required to select answers rather than generate them, and to provide concise responses to inquiries in which they have limited personal involvement [20]. On the other hand, "performance assessment" refers to forms of evaluation such as oral presentations, acting, and debates that allow students to demonstrate their knowledge in various ways. According to Stiggins [12], paper and pencil-based assessments can encompass teacher-made tests, quizzes, homework, seatwork exercises and assignments, observations of students' behaviours and evaluations of their work products for performance assessment.

## Teachers' practices in implementing classroom-based assessment

In academic environments, assessment is essential because it provides a comprehensive understanding of students' and teachers' learning progress [21]. To evaluate students' progress or focus on high-quality assessment, teachers must employ various assessment techniques in the classroom. Comprehending teachers' viewpoints and the rationale behind using different evaluation methods is vital for understanding their assessment approach. Black & Wiliam [22] and Stiggins [23] assert that teachers must have a strong foundation in classroom assessment techniques to establish a consistent method for both assessment for learning and assessment of learning.

Formative assessment, also called assessment for learning, fosters student learning and academic growth [24–26]. Van der Kleij et al. [27] noted that multiple terms describe formative assessment, and they investigated various assessment types for evaluating student learning, including monitoring, diagnostic, and formative assessments. Alternatively, summative assessment, known as assessment of learning, is executed at the end of a study period and helps make informed decisions. It offers an extensive summary of what was taught and learned throughout the session [28].

Classroom-based assessment perspectives and practices among teachers may need to be more consistent. Buyukkarci [29] studied the assessment beliefs and practices of language teachers in Turkish primary schools and discovered that although teachers supported formative assessment and feedback, they often needed to apply them consistently. Likewise, Saeed et al. [21] investigated teachers' views on incorporating classroom assessment techniques in primary and secondary schools in Lahore. Many public and private school teachers predominantly used summative assessment, but they also recognised the possible role of formative and summative assessments in enhancing student learning in the classroom.

Effective classroom-based assessment practices also emphasise the active participation of teachers, students, and their peers [30]. This underlines the importance of viewing students as information sources for themselves and their classmates [31]. Classroom-based assessment

that is more student-centred and encourages social learning processes, where students actively participate in the teaching and learning process, is deemed more effective than teacher-dominated methods [32].

Moreover, Harris & Brown [33] maintain that assessment results can be utilised for accountability purposes concerning students, teachers, and schools. Specifically, evidence from classroom assessments can evaluate the effectiveness of schools (school accountability), teaching quality (teacher accountability), and student learning outcomes (student accountability). It is important to note that accountability assessments are typically more connected with summative assessments than formative assessment practices [31].

## Classroom-based assessment practice in Malaysian ESL secondary school

The primary focus of the Malaysian CBA system is data collection and analysis to encourage consistent reflection on teaching and learning (T&L) and assist T&L improvement. According to KPM [1], CBA aims to help teachers, parents, and students discover areas for improvement in student learning, not to foster competition or comparisons among students. Teachers must actively participate in implementing CBA by developing learning objectives, creating assessment instruments, administering assessments, documenting outcomes, analysing data, and carrying out follow-up procedures. CBA seeks to advance academic attainment and learner mastery.

In Malaysia, classroom-based assessment (CBA) has been widely implemented in English language teaching to evaluate students' learning progress and provide feedback to teachers for instructional improvement. According to Santos et al. [34], CBA is a formative assessment that enables teachers to identify students' strengths and weaknesses in language skills and adjust their teaching strategies accordingly. There has been a growing interest in CBA as a practical approach to enhancing students' English proficiency in Malaysian secondary schools in recent years.

One of the critical benefits of CBA is its potential to improve students' learning outcomes. According to a study by Baghoussi [35], CBA helped promote students' active participation in classroom activities and encouraged them to take responsibility for their learning. The study found that when combined with teacher feedback, CBA resulted in significant improvements in students' language skills, particularly in reading and writing. However, implementing CBA in Malaysian secondary schools is challenging. In their study, Mohamad Marzaini et al. [36] identified several issues that could have improved the effective implementation of CBA, such as inadequate teacher training, lack of time and resources, and resistance from students and parents. These challenges highlight the need for a systematic and well-planned approach to CBA implementation in order to achieve its intended benefits.

In order to address these challenges, some researchers have suggested using technology to support CBA in English language teaching. For instance, Da-Hong et al. [37] proposed using an online platform to facilitate CBA, which allowed teachers to monitor students' progress and provide timely feedback. The study found that the platform enhanced students' engagement in learning and facilitated feedback sharing among teachers and students. CBA can potentially enhance students' English language proficiency in Malaysian secondary schools. However, the successful implementation of CBA requires adequate teacher training, appropriate resources, and a systematic approach that considers the challenges and opportunities CBA presents.

## The concept of pedagogical content knowledge

Shulman [38] is credited with introducing the concept of PCK (Pedagogical Content Knowledge). This framework recognises the importance of integrating general PCK, PK, and CK and represents a valuable tool for educational research (Fig 1). The Shulman model encompasses multiple domains of knowledge related to pedagogy, including instructional strategies, child

development, motivation, student needs and behaviour, and understanding of subject matter. The knowledge requirements for educators have undergone significant evolution throughout the history of education. For example, in the 1500s, Paris University required teachers to possess a high degree of knowledge. By the mid-1980s, teacher training and education programs had emphasised pedagogy more, with content knowledge as a secondary consideration [38].

Shulman [38] introduced the concept of PCK, which calls for integrating pedagogical and subject matter knowledge in teaching. He suggested that treating teaching methodology and content as separate entities is inadequate and that a more comprehensive approach is required to consider the specific needs and requirements of various curriculum areas. Over the years, the PCK framework has gained wide recognition and is frequently cited in educational research and literature. Shulman introduced the concept of PCK in his keynote address to the American Educational Research Association in 1985, which was subsequently published in 1986. Initially, PCK was regarded as a subset of content knowledge, aligned with Shulman's perspective on the significance of subject knowledge in education. Shulman's PCK framework was listed as one of seven components of a professional knowledge base for educational practitioners in 1987 by delegates who further explored his ideas. This model is a crucial foundation for substantial academic research in education and is considered an essential paradigm for educational scholars and practitioners.

## The relationship between teachers' pedagogical content knowledge and classroom assessment practices

Teachers' knowledge is considered 'learning specialist', which stands on how their knowledge functions and can be applied in their teaching and learning fraternity. In the context of classroom assessment practices, the knowledge encultured by the teachers is used in the process of making decisions pertaining to lesson design and making on-the-spot judgments in the classroom [39]. Ma'rufi et al. [40] affirmed that content and pedagogical knowledge have been closely related factors that affect teachers' assessment practices at the micro level. Shulman [38] posited that the PCK, PK, and CK are the three knowledge bases that combine to generate teachers' PCK.

In the practices of classroom-based assessment, the teachers' PCK has exceptionally played varying perspectives of teachers on how specific themes or subject matters are arranged, presented, and suited to the wide range of interests and abilities of the students might have an impact on how teachers implement classroom assessment [38]. Shulman also posited that PCK was the best fit suggested knowledge base for teaching, learning, and assessing because it serves as;

*The ability of a teacher to translate the topic knowledge they possess into pedagogically effective forms yet adaptable to the variances of evaluation in ability and background provided by the pupils is the key to differentiating their knowledge base of teaching.*

According to this definition, PCK includes understanding and pedagogical reasoning, transformation, and practice [38], which includes both teachers' understanding and their enactment of classroom assessment. Gess-Newsome [41] stated that teachers' PCK is a fundamental knowledge used in the preparation and delivery of teaching on a particular subject [42] and is also known as knowledge-on-action [43]. This knowledge is made explicit and available from teachers or places like lesson plans. Enactment or knowledge-on-action directs what the teacher does in the classroom, but it also involves the teacher making judgments immediately, requiring a more dynamic knowledge called knowledge-in-action [43].

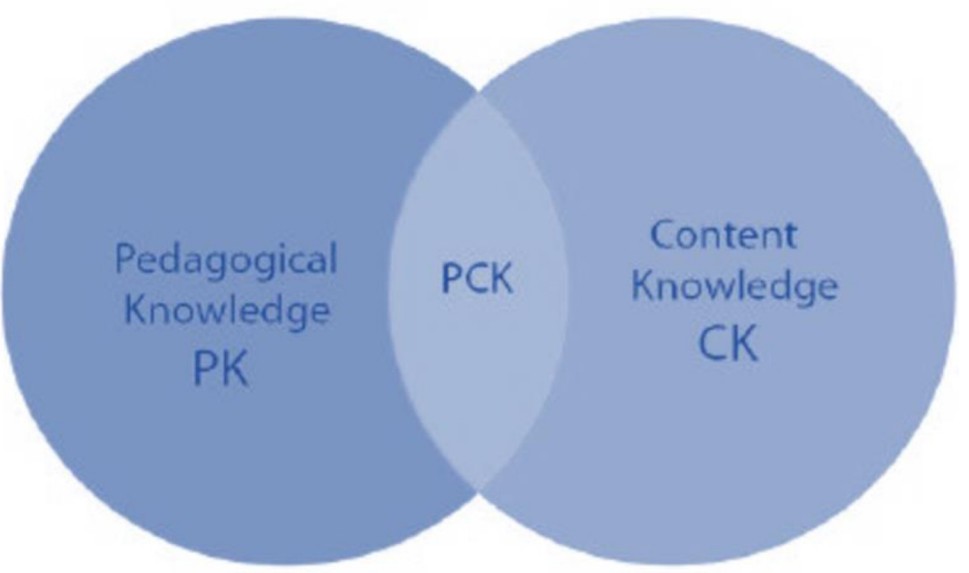

**Fig 1. Pedagogical content knowledge [38].**

Reflection-in-action during instruction allows for developing this knowledge-in-action [44]. This demonstrates the inverse (non-linear) relationship between teachers' PCK development and PCK implementation. Although sometimes clear, knowledge-in-action is typically implicit and more challenging to record [43]. According to this viewpoint, the relationship between knowledge and conduct is by its very nature reciprocal, as information is developed through experiences gained through classroom practice and interactions with peers [45]. Hence, this can contribute to the teachers' learning trajectories, and the teachers' PCK influenced the different ways of teachers' assessment practices.

## ESL teachers' content knowledge and classroom-based assessment practices

The relationship between English as a Second Language (ESL) teachers' content knowledge and classroom-based Assessment (CBA) has been a critical area of study in language education. Content knowledge, referring to teachers' understanding of the subject matter they teach, is essential in informing the design and implementation of effective classroom assessment practices. Researchers like Shulman [38] and Minor et al. [46] have emphasised the importance of content knowledge in shaping pedagogical practices, while other assessment researchers like Black & Wiliam [22], Puad & Ashton [31], and Liontou [47] have stressed the value of CBA as an integral part of the teaching and learning process. In the context of ESL teaching, the literature has explored how content knowledge influences assessment strategies, with scholars such as Jung Youn [48], Phothongsunan [49], and Savić [50] providing insights into the connection between language teachers' expertise and the development of context-sensitive, authentic assessments. This body of research demonstrates that a strong foundation in content knowledge can contribute significantly to the effectiveness of CBA in ESL settings, enabling teachers to design assessments that cater to the diverse needs of their students and ultimately enhance learning outcomes.

## ESL teachers' pedagogical knowledge and classroom-based assessment practices

Pedagogical knowledge encompasses the knowledge of teaching methods, instructional strategies, and learning theories that inform the instructional practices of ESL teachers. Scholars such as Shulman [38] and Borko & Putnam [51] have explored the importance of pedagogical knowledge in shaping teaching practices, while assessment experts like Black & Wiliam [22] have highlighted the significance of CBA in providing valuable feedback and promoting student learning. Within the context of ESL teaching, research conducted by Mohamad Marzaini et al. [52], Mat Yusoff et al. [5], and Dadvand & Behzadpoor [53] have shed light on the relationship between ESL teachers' pedagogical knowledge and the design, implementation, and interpretation of assessment in the classroom. These studies suggest that a solid foundation in pedagogical knowledge can enhance the effectiveness of CBA by informing the development of context-appropriate, learner-centred assessment strategies. Consequently, the literature underscores the need for continuous professional development to strengthen ESL teachers' pedagogical knowledge and optimise their use of CBA in diverse learning contexts.

## ESL teachers' pedagogical content knowledge and classroom-based assessment practices

In the realm of English as a Second Language (ESL) education, pedagogical content knowledge (PCK) and classroom-based Assessment (CBA) have emerged as critical factors in effective language teaching. Researchers such as Shulman [38], Kind & Chan [54], and Nind [55] have contributed significantly to the understanding of PCK, which refers to the specialised knowledge teachers possess, including their understanding of how to represent and teach a subject to diverse learners. Conversely, CBA involves teachers' implementation of assessment strategies within the classroom context to measure and enhance students' learning outcomes. Researchers like F Mudin [56], Abeywickrama [57], and Mohamad Marzaini et al. [7] have illuminated the importance of CBA in ESL settings, emphasising that authentic, formative assessment practices can significantly benefit student learning. Throughout the literature, the integration of PCK and CBA is recognised as a vital component in developing effective, learner-centred ESL teaching practices. Studies by Mumamad Marzaini et al. [52] and Ducar [58], for instance, highlight the intricate relationship between these two elements and suggest that further research is necessary to optimise the alignment of PCK and CBA in ESL contexts. Therefore, the hypotheses of the study can be posited as:

H1: *"There is a significant relationship between content knowledge and classroom-based assessment practices."*

H2: *"There is a significant relationship between pedagogical knowledge and classroom-based assessment."*

H3: *"There is a significant relationship between pedagogical content knowledge and classroom-based assessment."*

## Methodology

### Research design

The primary objective of this quantitative study, employing a survey research design, is to assess teachers' pedagogical content knowledge and classroom-based assessment practices. The researcher adopted a quantitative approach, explicitly utilising a survey research design, to comprehensively understand the educational context, particularly concerning teachers' perceptions, beliefs, and behaviours. Survey research is a widely used method in social sciences area, enabling researchers to systematically collect information related to human perceptions, beliefs, and behaviours [59]. Consequently, this method is well-suited for the present study, effectively addressing the research objectives. The quantitative approach allows for the measurement of variables. It facilitates in identifying the relationships and patterns among them, providing a robust understanding of the factors impacting pedagogical content knowledge and classroom-based assessment practices. By employing a survey research design, this study has the potential to uncover valuable insights into the intricacies of pedagogical content knowledge and assessment practices in the educational context, which can, in turn, contribute to the development of practical teacher training programs and strategies. Ultimately, this method can support the broader goal of enhancing educational outcomes by shedding light on the crucial role of teachers' pedagogical content knowledge and classroom-based assessment practices in the teaching and learning process.

### Sample

This study employed a simple random sampling method to ensure a representative sample of ESL teachers in secondary schools in Selangor, Malaysia, from January 2023 to April 2023. The researchers upheld ethical standards by securing permission letters for distributing the survey instrument and obtaining approvals from both the Educational Policy Planning and Research Division of the Ministry of Education Malaysia (approval letter number KPM.600-3/2/3-eras (13277)) and the University Malaya Research Ethics Committee (Non-medical) (approval letter number UM.TNC 2/UMREC). Prior to participating, written consent was obtained from all respondents.

To ensure the generalizability of research findings to the broader population, a widely recognized probability sampling technique was utilized, affording every member of the target population an equal opportunity for selection. This approach minimizes selection bias. The sample was drawn from 27 public secondary schools located in the Petaling Utama district of Selangor, encompassing 338 secondary school teachers. The gender distribution among the participants included 73 males and 265 females. Teaching experience varied from 1 year to more than 20 years, and participants' academic qualifications are detailed in Table 1. This diverse sample allows for a comprehensive exploration of teaching strategies employed by ESL teachers with diverse backgrounds and levels of experience.

### Instrumentation

For this study, a single survey was employed to collect data, adapted from Goggin [60], emphasising classroom-based assessment practices. The survey was developed by referencing "Classroom Assessment for Student Learning" by Chappuis et al. [61]. Five individuals, including four secondary school teachers and one teacher educator from University A, were consulted based on their experience with classroom-based assessment to ensure face validity. To establish content validity, the survey was distributed to twenty experts in the field through email, Researchgate, and social media platforms such as WhatsApp, Telegram, and Facebook

Table 1. Demographic information of the participants.

| Gender | Frequency |
|---|---|
| Male | 73 |
| Female | 265 |
| **Total** | **338** |
| **Years of Teaching** | |
| 1–5 years | 152 |
| 6–10 years | 89 |
| 11–15 years | 33 |
| 16–20 years | 24 |
| More than 20 years | 40 |
| **Total** | **338** |
| **Academic Qualifications** | |
| Diploma | 1 |
| Degree | 324 |
| Master's Degree | 13 |
| **Total** | **338** |

Messenger. The experts were selected based on their research experience, educational background, and accomplishments in related research activities [62]. Of the twenty experts approached, ten agreed to participate, two declined due to time constraints, and the remainder have yet to respond. As suggested by Hong et al. [63], a sample of experts with experience in the research area can enhance content validity and minimise biases in the research instrument. The experts were asked to evaluate the instrument's relevance, clarity, and simplicity by completing a questionnaire. This process helped ensure that the survey used in the study was both reliable and valid, thus contributing to the overall quality and rigour of the research findings.

## Data analysis

The researchers actively conducted screening tests in SPSS to address issues related to missing values, straightlining, and outliers. The researchers assessed missing values by implementing the count blank method. To ensure compliance with the straightlining rule, the researchers inspected for straightlining, noting a non-zero standard deviation value, as described by Hair et al. [64]. The team also performed an outlier test using the Mahalanobis Distance. After completing these steps, the researchers proceeded with a descriptive analysis.

The researchers used PLS-SEM for additional analysis, employing the SmartPLS software. This method maximises the explained variance of exogenous variables on endogenous ones to focus on prediction, as explained by Hair et al. [64] and Rigdon et al. [65]. Because it is suitable for testing theoretically supported and additive causal models, as Chin [66] and Haenlein & Kaplan [67] suggest, the team used it in line with the proposed theoretical framework, which is an exploratory model. Contrastingly, CB SEM tests the alignment between the proposed theoretical model and the observed data, as per Ramayah et al. (2018). It is, therefore, effective when working with confirmatory models. Wold [68] initially developed SEM, and his two disciples, Lohmoller and Joreskog, refined it. Lohmoller adapted it into PLS-SEM, while Joreskog developed it into CB-SEM.

Before progressing with hypothesis testing, the researchers verified internal consistency and convergent and discriminant validity. The researchers applied internal consistency to highlight the interrelation among items, as per Sekaran & Bougie [69], ensuring a composite reliability value of 0.7 for its successful implementation [64, 69]. The researchers examined convergent

validity through factor loadings and average variance extracted (AVE), demonstrating the degree to which specific indicators of a construct share a substantial proportion of variance, following Hair et al. [64]. They achieved convergent validity when factor loading values were at least 0.5 while also maintaining an AVE score above 0.5, according to Byrne [70]. The researchers also confirmed discriminant validity, demonstrating that each latent variable is distinct from others, pointing to the uniqueness of constructs [64, 71]. The researchers checked discriminant validity using the advanced method of the heterotrait-monotrait ratio of correlations (HTMT), with a threshold value of 0.9, as suggested by Franke & Sarstedt [72]. The researchers then conducted a lateral collinearity assessment with a cut-off value of 3.3 to prevent misleading results, based on Diamantopoulos & Siguaw [73]. Following these validation steps, the team performed hypothesis tests using structural modelling.

## Results

### The measurement model

The structural equation modelling (SEM) technique, which links measurement items to their underlying latent variables, necessitates establishing a measurement model. The present study provides the theoretical foundations and statistical analysis to support the validity and reliability of the measurement model. The analysis used the R programming language, utilising the lavaan package for SEM analysis [74].

### Operationalisation of latent variable

Four complex latent variables, which a single observed variable cannot assess, make up the conceptual model. In order to measure each of the latent variables in the conceptual model, the researchers use several observed items.

The first three exogenous variables relate to the teacher's PCK. The well-known study by Shulman [38] developed measurement scales to measure PCK. Under pedagogical content knowledge, there were 13 items divided into three dimensions. The items were adopted from past literature: CK [75, 76], PK [77], and PCK [77].

To measure the CBA practices, the researchers refer to related work from Chappuis et al. [61] and the items adopted from Goggin [60].

### Normality check

The researchers use multivariate and univariate normality tests to determine whether the measurement items are normal before moving forward with the confirmatory factor analysis (CFA) model. The estimating method used in CFA (and SEM) depends on the normality of the data. Hence, the data must be normal. The Shapiro-Wilk test (all p-values 0.05) of all measurement items rejects the null hypothesis of univariate normality, just as the Mardia test (p-value 0.05) does the same for the null hypothesis of multivariate normality. In order to estimate the measurement model, the researchers instead employ the maximum likelihood robust (MLR) estimator, also known as the Satorra-Bentler rescaling approach [74].

### Reliability and validity

Self-reported questionnaires are used to gather the measurement item data. Each item is scored on a 5-point Likert scale, with 5 denoting strong agreement and 1 denoting strong disagreement with the statements. The exploratory factor analysis (EFA) of the 21 items indicates a four-factor model after eliminating item one of the pedagogical knowledge construct and items one through two, five, seven, thirteen, and fourteen of the classroom assessment

construct (see Table 2). This model was later confirmed using confirmatory factor analysis (CFA). The right column of Table 2 displays the standardised factor loadings of the CFA model. All of them are statistically significant (p-value < 0.001), indicating that the items properly reflect the underlying latent construct. This attests to the measurement model's convergence validity (Anderson & Gerbing, 1988). Additionally shown in Table 2 are each factor's composite reliability (CR) and Cronbach's alpha [78]. According to Hair et al. [79], all factors' Cronbach's Alpha and CR values must be higher than the specified cutoff point of 0.70. Referring to Table 2, although pedagogical content knowledge and content knowledge are not exceeding 0.70 (moderate Cronbach's alpha), the researchers decided to maintain the construct as Pallant [80] said if the items are less than 10 in a construct, it is hard to get a high value of Cronbach's alpha, so it is acceptable if the value is >.05. On the other hand, high composite reliability and average variance extracted (AVE), and a moderate to high Cronbach's alpha, indicate a good measurement model in partial least squares structural equation modelling (PLS-SEM) [79, 81]. In this case, the researcher decided not to delete the item. Thus, the measurement model's reliability is confirmed.

**Table 2. Measurement items and their reliability.**

| Constructs and items | Factor loadings |
| --- | --- |
| Content Knowledge (CK, alpha: 0.63, CR: 0.95) | |
| 1. CK1 | .753 |
| 2. CK2 | .703 |
| 3. CK3 | .807 |
| Pedagogical Knowledge (PK, alpha: 0.81, CR: 0.87) | .668 |
| 1. PK1 | Dropped |
| 2. PK2 | .744 |
| 3. PK3 | .786 |
| 4. PK4 | .737 |
| 5. PK5 | .706 |
| 6. PK6 | .774 |
| 7. PK7 | Dropped |
| Pedagogical Content Knowledge (PCK, alpha: 0.59, CR: 0.83) | |
| 1. PCK1 | Dropped |
| 2. PCK2 | .826 |
| 3. PCK3 | .732 |
| Classroom-based Assessment (CBA, alpha: 0.94, CR: 0.95) | |
| 1. CBA1 | Dropped |
| 2. CBA2 | Dropped |
| 3. CBA3 | .852 |
| 4. CBA4 | .853 |
| 5. CBA5 | .793 |
| 6. CBA6 | .744 |
| 7. CBA7 | Dropped |
| 8. CBA8 | .786 |
| 9. CBA9 | .793 |
| 10. CBA10 | .760 |
| 11. CBA11 | .793 |
| 12. CBA12 | .797 |
| 13. CBA13 | .722 |
| 14. CBA14 | .752 |

If constructs that should not be related to one another are truly unrelated, this is known as divergent or discriminant validity (DV). This can be confirmed by contrasting the average extracted variance (AVE) with the squared correlations of all latent variables in a matrix, as illustrated in Table 3. According to Hair et al. [79], squared correlations below the diagonal must be less than the AVE of each latent variable to indicate DV. Table 4 shows that all constructs for AVE and squared correlation are below thresholds, confirming the DV of the latent variables. The researchers can thus attest to the latent variables' DV.

The Root Mean Square Error Approximation (RMSEA) and Standardized Root Mean Square Residual (SRMR), which are below the cut-off value of 0.08, as well as the Comparative Fit Index (CFI) and Tucker-Lewis Index (TLI), which are above the recommended threshold of 0.90 [79], more evidence for the measurement model's accuracy and a good fit (refer to Table 2). After establishing the measurement model, the researchers present the descriptive statistics in Table 4.

It should be noted that the mean and standard deviation (SD) values are determined by taking the arithmetic mean of the item scores that measure the respective latent variables. Based on factor values retrieved by confirmatory factor analysis (CFA), the correlation matrix shows a correlation between the latent variables.

## Assess the structural model for collinearity issue

The optimal VIF values, according to contemporary theories [82], should be lower than three (<3). For the PLS process, this study used Hair et al.'s [82] threshold of VIF values of less than three to prevent collinearity and common method bias. Knock [83] posits that a VIF exceeding 3.3 signifies excessive collinearity, potentially indicating the presence of common method bias within the model. Consequently, if all VIFs derived from a thorough collinearity assessment are at or below 3.3, the model can be deemed devoid of common method bias. The SmartPLS specifically generated the order of predictors to evaluate the collinearity. CK, PK, and PCK are predictors of CBA. The researchers assessed all VIF values; however, one indicator, PK (3.390), had values greater than 3. While the rest were assessed with the VIF values (<3), CK (2.123), and PCK (2.081). After considering the VIF values below, the researcher considered the work from Ringle et al. [84] that concur collinearity issues are performed when the VIF value is equal to or greater than five. Therefore, VIF values >3.3 and <5 were accepted in this study. According to the results of all VIF values (Table 5), it can be noted that collinearity is not an issue in this study.

Fig 2 and Table 6 show a structural equation model illustrating the relationships between Content Knowledge, Pedagogical Knowledge, and Pedagogical Content Knowledge with Classroom-based Assessment. The summary of hypothesis testing reveals that only Pedagogical Knowledge and Pedagogical Content Knowledge significantly influence Classroom-based Assessment.

**Table 3. Divergent validity analysis.**

|  | CBA | CK | PCK | PK |
|---|---|---|---|---|
| CBA | 0.787 |  |  |  |
| CK | 0.494 | 0.756 |  |  |
| PCK | 0.624 | 0.526 | 0.842 |  |
| PK | 0.702 | 0.693 | 0.737 | 0.750 |
| AVE | 0.657 | 0.618 | 0.748 | 0.601 |

**Table 4. Descriptive statistics and association among latent variables.**

| Latent Variables | Mean | SD | CBA | CK | PK | PCK |
|---|---|---|---|---|---|---|
| CBA | 3.80 | 0.58 | 1.00 | | | |
| CK | 4.06 | 0.43 | 0.49 | 1.00 | | |
| PK | 3.97 | 0.44 | 0.65 | 0.69 | 1.00 | |
| PCK | 3.91 | 0.46 | 0.62 | 0.52 | 0.73 | 1.00 |

## Common method bias

Measurement inaccuracies resulting from methodological problems are referred to as common method bias. For example, using the same measurement scale (e.g., a 5-point Likert scale) for all survey questions may result in common method bias. Podsakoff et al. [85] outline several statistical treatments for common method bias, each with advantages and disadvantages. Harman's single-factor test, which is the most commonly used, is used in this study. The researchers used the 21 items loading on one latent factor to perform unrotated exploratory factor analysis. Only 39% of the average variance is explained by a single factor, significantly less than the recommended cutoff point of 50%. So, in this study, common method bias is not a problem.

## Assess the level of R2

The coefficient of determination (R2) measures the proportion of variance in the dependent variable explained by the model's independent variables. In Partial Least Squares Structural Equation Modeling (PLS-SEM), an R2 value of 0.8 or greater is considered an acceptable fit of the model to the data, while an R2 value of less than 0.5 indicates a poor fit. In Table 7, the calculation of the coefficient determination (R2) is presented. The coefficient of determination (R2) score for the Classroom-Based Assessment (CBA) was found to be 0.472, reflecting a moderate level of explained variance of the dependent variable by the model. This indicates a satisfactory fit of the model to the data and a significant degree of accuracy in its prediction.

## Assess the f2 effect size

The magnitude of the f2 value is indicative of the strength of the relationship between the latent variable and its indicators. A higher f2 value implies a robust connection between these two elements and suggests that the latent variable effectively captures the assessed underlying construct.

According to Hair et al. [82], small, medium, and large effect sizes are categorised as .02, .15, and .35, respectively. Following the hypotheses, Table 8 presents the values of f2 for the endogenous constructions and the exogenous constructs or predictors: CK, PK, and PCK. The f2 value for the relationship between CK and CBA is .005, which is considered to have a small effect size. Likewise, the f2 value for PK and CBA is .098, indicating a small effect size. The f2 value for the relationship between PCK and CBA, .030, also depicts a similarly small effect size.

**Table 5. VIF values.**

| | CK | PCK | PK | CBA |
|---|---|---|---|---|
| **CK** | | | | 2.123 |
| **PCK** | | | | 2.081 |
| **PK** | | | | 3.390 |
| **CBA** | | | | |

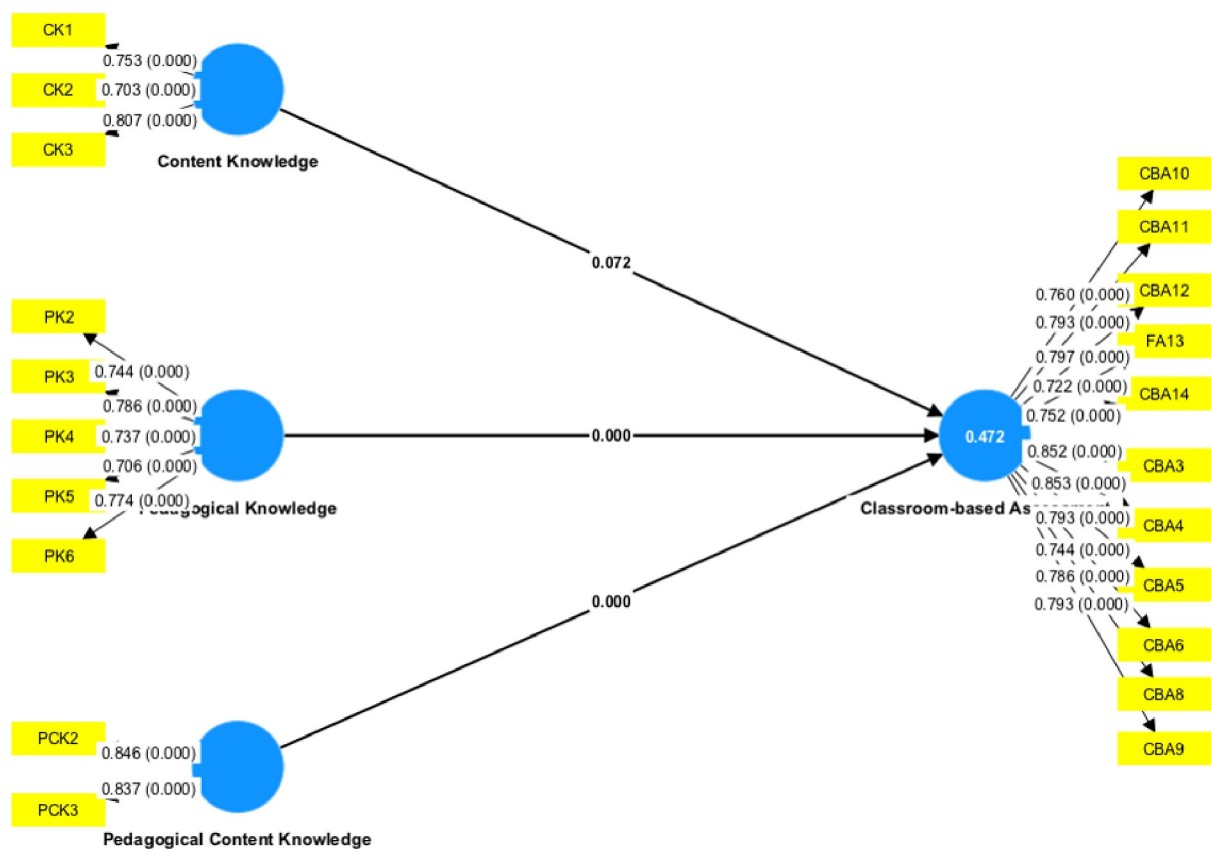

**Fig 2. The structural equation model was estimated.**

## Assessing the predictive relevance, $Q^2$

The predictive relevance of the model was assessed by utilising the Blindfolded Techniques in SmartPLS 4 [82] to obtain the Q2 value. The Q2 value measures a latent variable's capacity to predict the dependent variable in the model. A high Q2 value signifies that the latent variable holds a robust predictive ability towards the dependent variable, whereas a low Q2 value indicates a weak predictive ability. The results of the predictive relevance assessment, as presented in Table 9, show that the Q2 value of CBA is above 0 (.459), thereby validating the predictive validity of the endogenous construct in the model.

**Table 6. Summary of hypothesis testing.**

| Hypothesis | Standardised Coefficient | Remark |
|---|---|---|
| H1: Content Knowledge -> Classroom-based Assessment | 0.144* | Not supported |
| H2: Pedagogical Knowledge -> Classroom-based Assessment | 0.000** | supported |
| H3: Pedagogical Content Knowledge -> Classroom-based Assessment | 0.000** | supported |

Standard parenthesis error.

***$p < 0.001$,

**$p < 0.01$,

*$p < 0.05$,

+$p < 0.10$.

**Table 7. Results of R² of the integrated model.**

|  | $R^2$ | Consideration |
|---|---|---|
| CBA | .472 | Moderate |

## Discussion

Following establishing the measurement model in the previous sections, the researchers go on to the structural model to look at relationships between the latent variables. Once more, the researchers estimate SEM using the MLR method, as Rosseel [74] advised for non-normal data. In Fig 2, the researchers present the estimated SEM. It can be challenging to develop a theoretical structural model similar to the observed model at 5% statistical significance in complex SEM research with more than 12 measurement items, as in this study [79]. According to Bollen & Long [86], the chi-square statistic to degrees of freedom (DF) ratio in these circumstances should be less than three. The estimated SEM model (707.381/313 = 2.260) meets this requirement, indicating a robust model fit. Besides, other model-fit indices conditions are also satisfied. The RMSEA and SRMR are below 0.08, and the CFI and TLI are above 0.90. Hence, SEM estimations are thus valid. The model accounts for around 50% and 10% of the variance of the one endogenous variable, classroom-based assessment (as shown by the r-square of latent endogenous variables).

Other than that, the researchers summarise hypothesis testing in Table 5 based on the SEM model in Fig 2 and Table 5. All hypotheses relate to the association between teachers' knowledge and classroom-based assessment. Among those hypotheses, only H1 is not supported, suggesting that content knowledge is not associated with classroom-based assessment practices in the context of Malaysian secondary school teachers. In contrast, H2 and H3 are supported, suggesting a positive association between the pedagogy knowledge of teachers and classroom-based assessment practices. The overall results of path coefficient, t-value, p-values, $R^2$, $f^2$, and $Q^2$ can be seen in Tables 2–8 above.

Teachers' grasp of pedagogy holds significant importance since it can impact their practices in classroom-based assessment. This research focuses on investigating the pedagogical content knowledge and classroom-based assessment practices of secondary school teachers. In summary, the researchers first evaluate the PCK levels of secondary school teachers, which are based on their CK, PK, and PCK. As depicted in Table 4, the highest mean score is attributed to content knowledge (mean: 4.06, SD: 0.43), followed by PK (mean: 3.97, SD: 0.44), PCK (mean: 3.91, SD: 0.46), with classroom-based assessment (CBA) receiving the lowest but still relatively high mean score (mean: 3.80, SD: 0.58). Consequently, teachers may want to consider enrolling in additional courses that are relevant to their curriculum teaching, as this can enhance their confidence in PCK and, in turn, enable them to deliver an exceptional curriculum to their students more effective.

Similarly, a study done by Moh'd et al. [87], which assesses the level of teachers' PCK in selected secondary schools of Zanzibar, revealed that the level of teachers' PCK was moderate. According to the researchers, teachers' PCK plays a significant role in assessment practices

**Table 8. Results of f² of the integrated model.**

|  | $f^2$ | Effect Size |
|---|---|---|
| CK -> CBA | .005 | small |
| PK -> CBA | .084 | small |
| PCK -> CBA | .086 | small |

**Table 9. Results of Q2 of the integrated model.**

|  | $Q^2$ | Predictive relevance |
| --- | --- | --- |
| CBA | .459 | Large |

and students' performance, implying that teachers must have relevant knowledge in a particular area. Therefore, more in-service training on increasing teachers' PCK levels is required, ultimately resulting in better teaching and learning [87].

Next, the researchers delve into the impact of content knowledge on the classroom-based assessment practices of secondary school teachers. According to the findings, there is no discernible link between content knowledge and classroom assessment practices. This stands in contrast to the results obtained by Gess-Newsome et al. [42], who identified a substantial correlation between content knowledge and classroom assessment practices. Furthermore, these outcomes differ from those reported by Akram et al. [37], who established a noteworthy association between content knowledge and the utilization of ICT during teaching practice. One plausible explanation for these findings is that content knowledge may undergo occasional enhancements due to changes in the curriculum, leading teachers to perceive that it might not significantly influence their assessment practices [88]. Additionally, educators with high-quality content knowledge have the ability to seamlessly integrate subject matter expertise with other teaching knowledge, thereby facilitating comprehensive learning and assessment [40]. Akram et al. [6] underscore the significance of content knowledge, highlighting that a strong foundation in subject matter expertise is indispensable for effective online teaching. This implies that teachers' proficiency within their respective fields plays a pivotal role in their capacity to adapt teaching methodologies and engage students in meaningful learning experiences.

In addition, the researchers discovered a positive correlation between teachers' content knowledge and their assessment practises in the classroom. This result contradicts the findings of Herman et al. [89], who discovered no direct correlation between teachers' assessment use and their knowledge. According to the study, when teachers devote more time to evaluating and analysing student work, they gain valuable insights into how students' knowledge is developing and are able to identify any potential misunderstandings or obstacles. These findings are consistent with those of Akram et al. [6], who found that teachers in their study employed a variety of teaching strategies and techniques. The researchers emphasise the significance of diverse pedagogical approaches such as role modelling, narrative, and experiential activities, emphasising the significance of interactive and engaging teaching and learning techniques. Given the obvious relationship between pedagogical content knowledge and effective classroom assessment practises, it is recommended that teachers prioritise encouraging and continuously supporting student engagement with key subject concepts as part of their assessment practises in order to improve student learning [4].

Next, there is also a positive association between pedagogical knowledge and classroom assessment practices. This finding is in contrast with the findings from Depaepe & König [90], who found that there is no relationship between pedagogical knowledge and classroom assessment practices. It is generally believed that these two factors are connected and impact instructional practice [90]. Not only that, Akram et al. [6] found a crucial integration of pedagogical content knowledge in teaching and learning. The findings indicate that teachers exhibit positive perceptions regarding pedagogical content knowledge, believing that incorporating a solid understanding of subject matter and effective instructional strategies enhances teaching effectiveness and makes the learning process more exciting, interactive, and motivating for

students. Nevertheless, it is crucial to help educators develop pedagogical knowledge and facilitate its application in the classroom to assure high-quality instruction and favourably impact student learning outcomes [54].

## Conclusion

As a conclusion, this study has significantly contributed to the existing knowledge on ESL teachers' professional knowledge and assessment practices in Malaysian secondary schools. The findings indicate that ESL teachers possess a high level of professional knowledge, as demonstrated by their content knowledge, pedagogical knowledge, and pedagogical content knowledge. Furthermore, the study has identified positive correlations between pedagogical knowledge and classroom-based assessment and between pedagogical content knowledge and classroom-based assessment practices. In contrast, no relationship was found between content knowledge and classroom-based assessment.

In general, this study has the potential to enhance the professional development of ESL (English as a Second Language) teachers, which is an important contribution. By identifying positive correlations between pedagogical knowledge and pedagogical content knowledge and classroom-based assessment practises, the research highlights the significance of these specific domains in teacher preparation and ongoing professional development programmes. Educators and policymakers can use this information to devise targeted training modules that aim to improve teachers' assessment practises by enhancing their pedagogical knowledge.

In addition, the findings of the study have the implications for curriculum design and assessment guidelines as well. The identified correlations can be considered for incorporation into curriculum frameworks. For example, the curriculum can be structured to emphasise the development of pedagogical knowledge and pedagogical content knowledge, ensuring that teachers are well-equipped to implement classroom-based assessments. This, in turn, can result in curricula that are more comprehensive and student-centred. Moreover, the absence of a correlation between content knowledge and classroom-based evaluation is an intriguing observation. It suggests that effective ESL instruction requires a more holistic approach, rather than relying solely on content expertise. This finding challenges the conventional belief that content expertise alone is sufficient to make a teacher effective. Consequently, teacher training programmes can emphasise the significance of combining content knowledge and pedagogical expertise in order to develop well-rounded educators capable of optimising student learning outcomes.

As this study is in congruence with Akram et al.'s [6] emphasis on diverse pedagogical approaches such as role modelling, narrative, and experiential activities, this study highlights the importance of interactive and engaging teaching techniques. The findings call for a departure from traditional didactic teaching styles in favour of more interactive and participatory approaches, which can foster a deeper comprehension of the subject matter and improve assessment procedures. Finally, the study contributes to the academic field by emphasising areas in need of additional research. The lack of a direct relationship between content knowledge and classroom-based assessment, for instance, raises queries about the nuanced factors influencing ESL teacher effectiveness. Future research can delve deeper into these variables and investigate how they interact with teacher education and classroom dynamics. In conclusion, this study's contributions transcend the immediate context of ESL instruction in secondary institutions in Malaysia. They have the potential to inform pedagogical practises, curriculum development, and teacher training programmes, which will ultimately benefit both ESL teachers and students.

## Acknowledgments

The authors sincerely thank the numerous secondary school teachers who graciously participated in the study as volunteers.

## Author Contributions

**Conceptualization:** Rafiza Abdul Razak.

**Data curation:** Shahazwan Mat Yusoff, Anwar Farhan Mohamadd Marzaini.

**Formal analysis:** Shahazwan Mat Yusoff.

**Investigation:** Shahazwan Mat Yusoff, Anwar Farhan Mohamadd Marzaini.

**Methodology:** Shahazwan Mat Yusoff, Chin Hai Leng.

**Resources:** Chin Hai Leng.

**Supervision:** Rafiza Abdul Razak, Chin Hai Leng.

**Validation:** Chin Hai Leng, Anwar Farhan Mohamadd Marzaini.

**Writing – original draft:** Shahazwan Mat Yusoff.

**Writing – review & editing:** Rafiza Abdul Razak, Anwar Farhan Mohamadd Marzaini.

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
