## [Decision Letter · Decision Letter 0]

25 Jun 2023

PONE-D-23-14901Evaluating Teachers’ Pedagogical Content Knowledge in implementing Classroom-Based Assessment: A Case Study Among ESL Secondary School Teachers in Selangor, MalaysiaPLOS ONE

Dear Dr. Mat Yusoff,

Thank you for submitting your manuscript to PLOS ONE. After careful consideration, we feel that it has merit but does not fully meet PLOS ONE’s publication criteria as it currently stands. Therefore, we invite you to submit a revised version of the manuscript that addresses the points raised during the review process.

We look forward to receiving your revised manuscript.

Kind regards,

Ahmad Samed Al-Adwan

Academic Editor

PLOS ONE

4. Please amend the manuscript submission data (via Edit Submission) to include authors Chin Hai Leng and Anwar Farhan Mohd Marzaini.

Additional Editor Comments:

Dear authors,

Thank you for considering PLOS ONE. The reviewers have identified some major concerns that need to be addressed by you. Specifically, it is important to highlight the main contributions of your study in the introduction section. Furthermore, both practical and theoretical implications of your study. Full report of the reviewers' comments is attached in this e-mail.

Regards,

Reviewers' comments:

Reviewer's Responses to Questions

**Comments to the Author**

1. Is the manuscript technically sound, and do the data support the conclusions?

Reviewer #1: Yes

Reviewer #2: Yes

2. Has the statistical analysis been performed appropriately and rigorously? 

Reviewer #1: Yes

Reviewer #2: Yes

3. Have the authors made all data underlying the findings in their manuscript fully available?

Reviewer #1: Yes

Reviewer #2: Yes

4. Is the manuscript presented in an intelligible fashion and written in standard English?

Reviewer #1: Yes

Reviewer #2: Yes

5. Review Comments to the Author

Reviewer #1: thank you for considering PLOS for you submission. This an interesting topic that aim to evaluating teachers’ pedagogical content knowledge in implementing classroom based Assessment I do believe the submission have a merit. However, there are a few suggestions that may increase the quality if the this paper.

1. The authors are suggested to formulate explicit hypotheses of the proposed relationships in figure 2. Further, these hypotheses should be theoretically supported.

2. The discussion section can be enhanced by performing more comparisons between the findings of this study and the relevant literature.

3. In the methodology section, it would be useful to have a paragraph to outline the process of data analysis.

Reviewer #2: This is a well-written research paper that has good contribution in the education field, particularly in Malaysia. In order to increase the quality of this paper, several aspects should be considered.

1. The introduction section of this paper lacks clear presentation of its main contributions. The authors should emphasize the research problems, significance, and gaps in the existing literature to effectively highlight their contributions.

2. In order to provide a comprehensive and organized research report, it is essential to present both theoretical and practical contributions separately following the discussion section. This separation allows for a clear distinction between the concepts and ideas put forth in the theoretical framework and the tangible outcomes observed in the practical implementation.

3. While the analysis section includes the main elements of a good analysis, the multi-collinearity test is missed. Please perform and report.

4. The introduction, discussion, and the proposed hypotheses can be enhanced by including recent and relevant literature:

- Pedagogical practices and challenges in cultivating moral values: A qualitative study of primary school teachers in Pakistan. Doi: https://doi.org/10.1080/03004279.2021.1992471

- Beliefs and Knowledge for Pre-Service Teachers’ Technology Integration during Teaching Practice: An Extended Theory of Planned Behavior Doi: https://doi.org/10.1080/07380569.2022.2124752

- Teachers’ Perceptions of Technology Integration in Teaching-Learning Practices: A Systematic Review Doi: https://doi.org/10.3389%2Ffpsyg.2022.920317

- Technology Integration in Higher Education During COVID-19: An Assessment of Online Teaching Competencies Through Technological Pedagogical Content Knowledge Model Doi: https://doi.org/10.3389/fpsyg.2021.736522

6. PLOS authors have the option to publish the peer review history of their article (what does this mean?). If published, this will include your full peer review and any attached files.

Reviewer #1: No

Reviewer #2: No

---

## [Author Response · Author response to Decision Letter 0]

18 Jul 2023

Reviewer #1: thank you for considering PLOS for you submission. This an interesting topic that aim to evaluating teachers’ pedagogical content knowledge in implementing classroom based Assessment I do believe the submission have a merit. However, there are a few suggestions that may increase the quality if the this paper.

1. The authors are suggested to formulate explicit hypotheses of the proposed relationships in figure 2. Further, these hypotheses should be theoretically supported.

We acknowledge the importance of theoretically supported hypotheses and ensure that they have included in our manuscript.

2. The discussion section can be enhanced by performing more comparisons between the findings of this study and the relevant literature.

We agree that enhancing the discussion section with more comparisons between our findings and the relevant literature would be beneficial. We have further explored and incorporated these comparisons to provide a more comprehensive analysis in our manuscript.

3. In the methodology section, it would be useful to have a paragraph to outline the process of data analysis.

We agree that providing a clear explanation of our data analysis procedures will enhance the transparency and reproducibility of our study. We have included a detailed description of the data analysis process in the revised manuscript.

Reviewer #2: This is a well-written research paper that has good contribution in the education field, particularly in Malaysia. In order to increase the quality of this paper, several aspects should be considered.

1. The introduction section of this paper lacks clear presentation of its main contributions. The authors should emphasize the research problems, significance, and gaps in the existing literature to effectively highlight their contributions.

We acknowledge the importance of emphasizing the research problems, significance, and gaps in the existing literature to effectively highlight our contributions. In the revised manuscript, we have provided a more explicit and focused presentation of our main contributions to address this concern.

2. In order to provide a comprehensive and organized research report, it is essential to present both theoretical and practical contributions separately following the discussion section. This separation allows for a clear distinction between the concepts and ideas put forth in the theoretical framework and the tangible outcomes observed in the practical implementation.

In the revised manuscript, we have done the separation of theoretical and practical contributions to enhance the clarity and organization of our research report.

3. While the analysis section includes the main elements of a good analysis, the multi-collinearity test is missed. Please perform and report.

We agree that this test is important to assess the presence of collinearity among the variables. In the revised manuscript, we have conducted a multi-collinearity test and reported the results to ensure a comprehensive analysis.

4. The introduction, discussion, and the proposed hypotheses can be enhanced by including recent and relevant literature:

- Pedagogical practices and challenges in cultivating moral values: A qualitative study of primary school teachers in Pakistan. Doi: https://doi.org/10.1080/03004279.2021.1992471

- Beliefs and Knowledge for Pre-Service Teachers’ Technology Integration during Teaching Practice: An Extended Theory of Planned Behavior Doi: https://doi.org/10.1080/07380569.2022.2124752

- Teachers’ Perceptions of Technology Integration in Teaching-Learning Practices: A Systematic Review Doi: https://doi.org/10.3389%2Ffpsyg.2022.920317

- Technology Integration in Higher Education During COVID-19: An Assessment of Online Teaching Competencies Through Technological Pedagogical Content Knowledge Model Doi: https://doi.org/10.3389/fpsyg.2021.736522

We appreciate your suggestion to include recent and relevant literature in the introduction, discussion, and proposed hypotheses. We have carefully reviewed the suggested articles and integrated relevant insights from them into our manuscript. By incorporating these references, we aim to enhance the scholarly foundation of our study and establish connections with the existing literature.

---

## [Decision Letter · Decision Letter 1]

27 Sep 2023

PONE-D-23-14901R1Evaluating Teachers’ Pedagogical Content Knowledge in implementing Classroom-Based Assessment: A Case Study Among ESL Secondary School Teachers in Selangor, MalaysiaPLOS ONE

Dear Dr. Mat Yusoff,

Thank you for submitting your manuscript to PLOS ONE. After careful consideration, we feel that it has merit but does not fully meet PLOS ONE’s publication criteria as it currently stands. Therefore, we invite you to submit a revised version of the manuscript that addresses the points raised during the review process.

ACADEMIC EDITOR:  Your responses to the reviewers' concerns were clear and well-thought-out, and I'm pleased to see that you've incorporated their feedback into the revised version. This demonstrates your commitment to producing high-quality research and your willingness to engage constructively with peer review. However, I did notice that the paper still requires some proofreading. There are a few minor grammatical and typographical errors throughout the text that could be polished to enhance the overall readability of the paper. A careful proofreading pass or the assistance of a professional editor would be beneficial in this regard. Overall, I commend your dedication to improving your work and believe that with some final polishing, your paper will be in excellent shape for publication.

We look forward to receiving your revised manuscript.

Kind regards,

Ahmad Samed Al-Adwan

Academic Editor

PLOS ONE

Journal Requirements:

Additional Editor Comments:

Thank you for resubmitting the revised version of your paper. Your responses to the reviewers' concerns were clear and well-thought-out, and I'm pleased to see that you've incorporated their feedback into the revised version. This demonstrates your commitment to producing high-quality research and your willingness to engage constructively with peer review. However, I did notice that the paper still requires some proofreading. There are a few minor grammatical and typographical errors throughout the text that could be polished to enhance the overall readability of the paper. A careful proofreading pass or the assistance of a professional editor would be beneficial in this regard. Overall, I commend your dedication to improving your work and believe that with some final polishing, your paper will be in excellent shape for publication.

Reviewers' comments:

Reviewer's Responses to Questions

**Comments to the Author**

1. If the authors have adequately addressed your comments raised in a previous round of review and you feel that this manuscript is now acceptable for publication, you may indicate that here to bypass the “Comments to the Author” section, enter your conflict of interest statement in the “Confidential to Editor” section, and submit your "Accept" recommendation.

Reviewer #3: All comments have been addressed

Reviewer #4: All comments have been addressed

2. Is the manuscript technically sound, and do the data support the conclusions?

Reviewer #3: Yes

Reviewer #4: Yes

3. Has the statistical analysis been performed appropriately and rigorously? 

Reviewer #3: Yes

Reviewer #4: Yes

4. Have the authors made all data underlying the findings in their manuscript fully available?

Reviewer #3: Yes

Reviewer #4: Yes

5. Is the manuscript presented in an intelligible fashion and written in standard English?

Reviewer #3: Yes

Reviewer #4: Yes

6. Review Comments to the Author

Reviewer #3: Accepted as it is, Accepted as it is.Accepted as it is.Accepted as it is.Accepted as it is.Accepted as it is.Accepted as it is.Accepted as it is.Accepted as it is.Accepted as it is.Accepted as it is.Accepted as it is.Accepted as it is.Accepted as it is.Accepted as it is.Accepted as it is.Accepted as it is.Accepted as it is.Accepted as it is.Accepted as it is.Accepted as it is.Accepted as it is.Accepted as it is.Accepted as it is.Accepted as it is.Accepted as it is.Accepted as it is.Accepted as it is.Accepted as it is.Accepted as it is.Accepted as it is.Accepted as it is.Accepted as it is.Accepted as it is.Accepted as it is.

Reviewer #4: The manuscript presents an interesting study. However it is advised that authors send the manuscript for professional proofreading, because there are some inconsistencies in the essay, and language style used. There are also references in text which were not cited in the Reference section. Authors are advised to not make "sweeping statements" about the generalisability of the findings, because the study only looked at one state in Malaysia.

7. PLOS authors have the option to publish the peer review history of their article (what does this mean?). If published, this will include your full peer review and any attached files.

Reviewer #3: **Yes: **No objection

Reviewer #4: No

---

## [Author Response · Author response to Decision Letter 1]

3 Oct 2023

Journal Requirements:

If you need to cite a retracted article, indicate the article’s retracted status in the References list and also include a citation and full reference for the retraction notice.

We have clearly indicated the retracted status of the relevant articles in the References list of our manuscript. Additionally, we have provided citations and full references for the associated retraction notice as required.

Additional Editor Comments:

However, I did notice that the paper still requires some proofreading. There are a few minor grammatical and typographical errors throughout the text that could be polished to enhance the overall readability of the paper. A careful proofreading pass or the assistance of a professional editor would be beneficial in this regard.

We appreciate your suggestion to improve the readability of the manuscript through meticulous proofreading to correct minor grammatical and typographical errors present in the text. We have taken your advice to heart and have thoroughly proofread the entire manuscript. We have corrected all identified errors and have endeavoured to refine the language and presentation throughout to improve overall readability and coherence. We have also sought the assistance of a professional editor to ensure that no mistakes remain and to further polish the language. We believe these revisions have significantly improved the quality of the manuscript and have addressed the concerns you raised. We am hopeful that these modifications will meet your approval, and We are open to any further suggestions or corrections you may have.

Reviewer #4: 

The manuscript presents an interesting study. However it is advised that authors send the manuscript for professional proofreading, because there are some inconsistencies in the essay, and language style used. There are also references in text which were not cited in the Reference section. Authors are advised to not make "sweeping statements" about the generalisability of the findings, because the study only looked at one state in Malaysia.

Thank you very much for your insightful comments and constructive feedback on our manuscript. We would like to confirm that we have engaged the services of a professional editor to proofread the manuscript thoroughly and address inconsistencies in the essay and language style, ensuring that the manuscript meets the high standard of quality expected of academic publications. We have carefully reviewed the manuscript to ensure that all references mentioned in the text are accurately and completely cited in the Reference section. We apologize for any oversight in the initial submission and appreciate your patience as we rectified this. We acknowledge your concern regarding the generalizability of our findings, given that our study focused solely on one state in Malaysia. In light of your advice, we have revised the relevant sections of the manuscript to clarify the scope and limitations of our study and have refrained from making sweeping statements about the generalizability of our findings. We now explicitly state the contextual boundaries of our results and emphasize the need for caution when extrapolating the results to other settings. We believe that these revisions have addressed your concerns effectively and have enhanced the overall quality of the manuscript.

---

## [Editor Report · Decision Letter 2]

11 Oct 2023

Evaluating Teachers’ Pedagogical Content Knowledge in implementing Classroom-Based Assessment: A Case Study Among ESL Secondary School Teachers in Selangor, Malaysia

PONE-D-23-14901R2

Dear Dr. Mat Yusoff,

We’re pleased to inform you that your manuscript has been judged scientifically suitable for publication and will be formally accepted for publication once it meets all outstanding technical requirements.

Kind regards,

Ahmad Samed Al-Adwan

Academic Editor

PLOS ONE

Additional Editor Comments (optional):

Thank you for addressing the revised version of your paper. I can see that you have fully addressed the reviewers’ comments.
---

## [Editor Report · Acceptance letter]

18 Dec 2023

PONE-D-23-14901R2 

PLOS ONE

Dear Dr. Mat Yusoff, 

I'm pleased to inform you that your manuscript has been deemed suitable for publication in PLOS ONE. Congratulations! Your manuscript is now being handed over to our production team.

Kind regards, 

on behalf of

Professor Ahmad Samed Al-Adwan 

Academic Editor

PLOS ONE